# Challenges in the Adoption of eHealth and mHealth for Adult Mental Health Management—Evidence from Romania

**DOI:** 10.3390/ijerph19159172

**Published:** 2022-07-27

**Authors:** Andra Ioana Maria Tudor, Eliza Nichifor, Adriana Veronica Litră, Ioana Bianca Chițu, Tamara-Oana Brătucu, Gabriel Brătucu

**Affiliations:** 1Faculty of Economic Sciences and Business Administration, Transilvania University of Brașov, Colina Universității Street No. 1, Building A, 500068 Brașov, Romania; andra.tudor@unitbv.ro (A.I.M.T.); adriana.litra@unitbv.ro (A.V.L.); ioana.chitu@unitbv.ro (I.B.C.); gabriel.bratucu@unitbv.ro (G.B.); 2Faculty of Psychology and Educational Sciences, Transilvania University of Brașov, N. Bălcescu Street No. 56, 500019 Brașov, Romania; tamara.bratucu@cseibv.ro; 3The School Center for Inclusive Education Brasov, 125 Bd. 13 Decembrie, 500164 Brașov, Romania

**Keywords:** mental health management, eHealth, mHealth, digitalization for medical services, mental health, health needs

## Abstract

New methods of connecting physicians and patients have arisen. Technology is playing a crucial role and the concept of hybrid doctor–patient relationship is considered relevant for the competitive health management system. At the same time, the need for knowledge about implementing policies and best practices into the system is highly demanding. Digital tools, such as eHealth or mHealth can improve the traditional approach to consulting patients without requiring face-to-face interaction. However, due to the discussion surrounding the adoption of these technologies, the authors performed the study with two marketing research methods. The first is qualitative and is related to the opinions, attitudes, and beliefs of Romanian experts on the use of eHealth and mHealth for the prevention, detection, and treatment of mild mental disorders. The second method quantifies the opinions, attitudes, and behaviours of Romanian adults on their openness to adopt new technologies for mental health management. The main findings of the research highlight three factors that can increase the chances of adults using technology for health-related needs: (1) accessibility (2) data security, and (3) content. These are the main aspects that influence the well-being of both young and older adults, who both need support regarding mental health management.

## 1. Introduction

The concept of a hybrid doctor–patient relationship is presented currently as the best practice in mental health management [1]. The traditional approach to consulting patients faces to face is no longer among today’s requirements. Due to this, many aspects have changed, not only those related to ethics. The applications of technology in the mental health and society requires the adaptation to a new era, where new software, apps, or digital tools, such as eHealth and mHealth become usual. Society is going through a continuous technological revolution that makes it increasingly easier for individuals to access information and communication tools (hereinafter ICT) [2].

In the health sector, particularly during the COVID-19 pandemic, the boom in the use of ICT tools has manifested as an evolution among users, with a transition from acceptance of digital health to understanding and trusting of its potential and practicability [3]. Health decision-makers will need to address public health policies on the assumption that the health services market is constantly changing as new technologies are developing to help diagnose, treat, or even prevent various new and old diseases [4,5]. 

In this context, the need arises for the development of an integrated health management system as most of the time the communication strategies used in the healthcare market have focused on campaigns to prevent and raise awareness of the negative effects on the targeted public, also the changes in the pillars on which they are based [6,7].

The profile of the consumer is important on all markets in designing policies and strategies to address their needs. This also happens on the mental health market, and it is important in setting the proper mental health management decisions. Romania, where the research was conducted, lacks educational, prevention, and promotion policies implemented in the mental health area, and therefore, according to previous studies and statistics, depression is detected and treated in its advanced phases, which means increased costs for all parties (patient, healthcare providers, government, etc.), compared to the case of early-stage detection and treatment of depression.

eHealth and mHealth are still, unfortunately in early stages in Romania, and thus in the country, there were no previous studies on these concepts applied to mental health purposes. eHealth and mHealth comprise tools for each healthcare subdomains and action field; therefore, in the present, we cannot extend these to all eHealth and mHealth tools but rather to prevention and promotion, and self-care of mental health related issues.

In this context, the authors aimed to identify very accurately the attitudes and behaviours of Romanian experts regarding the use of eHealth and mHealth to prevent mild mental disorders, such as depression and anxiety, and to determine the profile of the Romanian consumer of health services.

The most important result was obtained by applying the binary logistic regression method, which revealed that there is a relationship between the determinants of the use of technology by adults and the intention to use the technology for health-related needs. The constructed model illustrated that accessibility, content, data security, communication options are important factors in the respondents’ decision to use technology for health-related needs, where, according to the tests conducted, accessibility can increase the chance of an adult using technology for health-related needs by more than 300%; data security increases the chance of an adult using technology for health-related needs by 94.1%, and content can increase the chance of an adult using technology for health-related needs by six times.

Healthcare in general and mental health are sensitive topics for patients; thus, the research results emphasize the importance of data security regarding such applications. Guaranteeing the security in communication, storage and transfer of personal and medical data are not only imposed by law but a request of users that once applied could make such tools trustworthy for them.

### 1.1. Literature Review

There are many definitions of eHealth and mHealth in the literature; however, many of them are similar, differing only in the detail of the concept or its placement in a particular context [8].

The concept of eHealth refers to medical practice supported by electronic and communication processes. While some specialists refer to eHealth as the equivalent of health informatics, covering all electronic or digital processes in healthcare [9], others consider eHealth as referring to healthcare services using the Internet [10,11]. eHealth is a widely used concept that encompasses a range of services or systems that are at the interface between healthcare and information technology.

This includes electronic health records—these allow patient data to be communicated between healthcare professionals; computerized scheduling—electronically requesting medical services; electronic prescribing—access to prescribing options, printing prescriptions to patients; electronic transmission of prescriptions into the medical system; providing digital information about protocols and standards for health professionals to use in diagnosing and treating patients [12]; telemedicine—remote diagnosis and treatment, including telemonitoring of patient functions; remote exchange of clinical information and images [13]; the use of electronic resources on health topics by healthy people or patients; electronic tools for health education management; virtual healthcare teams; and mHealth—reformed health and medical care achieved through mobile devices, in order to improve and support the prevention, diagnostics, therapy, monitoring, and follow-up care of patients [14].

The concept of mHealth refers to the use of media technologies and mobile telecommunications on a large scale for the distribution of health-related services and information. In the last decade, the definition of mHealth has expanded to the public health and wellness segment, and the concept of “mobile phone” no longer refers only to voicemail, SMS, or MMS but also to smart capabilities, such as Internet access, video, image functions, etc. 

Although smartphones are often inaccessible in poor countries, mHealth has the potential to revolutionize health information systems and generate individualized healthcare experiences for people around the world [15]. The adoption of mHealth technology in high-income countries is over 60% compared to 20% among middle and low-income countries; this significant difference is likely associated with better awareness and understanding of mHealth technology by the health systems of the high-income countries [16].

Differences between segments of mobile phone markets are not due to demographics but rather how people use phones. For example, adults aged 18–29 years old use mobile phones for SMS, IM, email, and the internet much more frequently than older people [17]. Mobile phones create a unique opportunity to overcome many psychological and social barriers that prevent users from accessing public health services (an intangible version of “cost”), even more so in the special conditions given by the pandemic [18,19,20]. On the other hand, the pandemic is precisely the situation that has made evident the failure of the effectiveness of digital health tools in poor or underdeveloped areas, although they are widely used in developed and developing countries [21].

The most obvious power of the mobile revolution in health is that the devices enable consistent and direct communication. Health-related information can be provided on demand to individuals whenever they want to access them. For example, the Mayo Clinic InTouch program offers a “Symptom Checker” for subscribers for a quick assessment of disease severity, along with a first aid guide with detailed tips on treating and responding to medical emergencies, an emergency locator to detect nearby medical facilities, health alerts, healthy living tips, and other benefits [22].

The mHealth revolution is only beginning, and new technologies are becoming a viable option as they develop. For example, considering that the telephone could become a vital component of emergency obstetric care, profile organizations (such as Maternova) are evaluating the effectiveness of technologies by testing them in the field by midwives [23]. Innovations demonstrate how mHealth is a continuously growing sector and teammate with marketing opportunities.

The most common application of mHealth is the use of mobile phones and communication devices to educate consumers about preventive healthcare services. However, mHealth apps are also used in various purposes, improving patient outcomes as well as the overall quality of healthcare provided [24]. Five major themes of mHealth applications can be defined [25]: treatment compliance referring to the use of mobile applications to ensure that patients strictly adhere to their treatment schedule [26,27]; data and disease monitoring [28,29,30]; creation of health information and support systems at the care place—whereas health information systems were previously created only for physicians and nurses, the eHealth and mHealth revolution brings changes to target groups [31]; health promotion and disease prevention [32]—SMS messages can be useful for distributing essential health education materials, as well as information about disease prevention and public clinic locations; and responding to medical emergencies [33,34]—this sphere is the most difficult to address, mainly due to the nature and trends of health emergencies. 

However, mobile technologies represent a potential tool that is useful in receiving timely healthcare in case of emergency. Barriers include constraints, such as network capacity and infrastructure costs (transport and roads).

Further research is needed in all five areas of healthcare as well as the continued development of e-Health and mHealth to overcome barriers within the current system. Although most of the results of the widespread use of technology devices reported by specialists have been positive (there are also cases where it is acknowledged that, although effective when applied, they sometimes raise the issue of low adherence, both in the treatment of psychiatric conditions [35] and the treatment of physical conditions [36,37], some negative repercussions may also occur (loss of data, viruses, system errors that may generate communication problems between users and medical organisations, sale of biometric data for other purposes, etc.).

Overall, the use of technology devices, and in particular mobile devices, among public health staff provides an opportunity for an enhanced and improved relationship with the target audience. The benefits of technology tools go beyond communication, as they can serve as marketing tools that can be found in every part of the marketing mix [38]. Technology devices have the potential to revolutionize social marketing in the public health sector, as they provide immediate access to health information, offer social support and connection and relationship building capabilities, effectively engage audiences, and help collect data and provide feedback.

### 1.2. Romania in the European Context on Digitization in Medicine

eHealth, mHealth, and telemedicine have become increasingly used terms in medical practice through electronic processes and communication. The European Commission has drawn up an action programme for health for the period 2014–2020, focused on improving the health of EU citizens and taking action to create sustainable health systems [39] Among the objectives of the European Commission’s programme for developing the eHealth sector are diversifying the unique integrated health information system; ensuring interoperability between health systems at the local, national, and European level; monitoring and controlling the health of EU citizens; improving preventive health actions; expanding and creating a variety of emergency health services; and developing eHealth, mHealth, and telemedicine solutions to facilitate prophylactic methods [40].

Romania has implemented a Unique Integrated System, a solution for the efficient management of the health insurance fund through the online collection and management of the medical information of the insured people. The health-specific IT systems that have been implemented thus far in Romania are “The Unique Integrated System”, “Classification System by Diagnostic Groups” (GDI), “Electronic Prescription System” (RE), “Health Insurance Card System” (SCAS), “Electronic Vaccination Register” (REV), as well as “Electronic Patient Record Management System” (SGEFP) and SMURD but also “Telemedicine System for Rural Areas” [41].

In Romania, there is no authority coordinating eHealth policy, and the Ministry of Public Health is responsible for all eHealth activities at the national level. Regarding the eHealth infrastructure, 97% of GP practices use a computer during a medical consultation [42], and of GPs, about 65% store patients’ medical records electronically [43]. Even so, the implementation of health data exchange is low, with only 19% of GPs using health data exchange compared to the EU average of 43% [44].

The Digital Single Market strategy focuses on e-health, mHealth, and telemedicine services [45] Figure 1 illustrates the value of the global eHealth market in 2019 and its projected value in 2026, with experts estimating that the value of the market will increase six-fold in 2026 compared to 2019 [46]. At the same time, the market for mHealth services and applications is expected to pass 40% coverage globally [47].

The percentage of the population searching the internet for medical information, the European average is 56%, and Romania ranks penultimate with 40%, ahead of Bulgaria (36%), with Finland being the country where most people accessed the internet for medical information (80%) [48].

According to Eurostat data, in Romania, the percentage of individuals who accessed the internet for medical information has evolved from 11% in 2008 to 33% in 2017. Interestingly, in 2018, the percentage of the population that accessed the internet for medical information decreased to 31%. This percentage was maintained throughout 2019. This decrease may be due to the phenomenon of migration of young people from Romania to other countries, this phenomenon being an alarming one during the mentioned period [48].

Although the use of medical data exchange is quite low, Romania ranks 16th in the ranking of prescription services, with 29% of the medical staff using these services, closer to the EU average of 50% in 2018. As for eHealth services, according to 2017 data, Romania ranks 21st among EU countries in the implementation of these services, with a rate of 11%, below the EU average of 18% in the same year [44].

Technological progress, as well as the steps taken to digitize healthcare services, are influencing the performance of health systems. New technologies offer new methods and techniques to determine consumer needs and improve health services. Even so, the results of digitization in the healthcare market depend on the quality of the process, the involvement of all parties (both consumers and providers), the involvement of IT specialists but also governments and the responsible central authorities [49].

Digitized health services can contribute to the sustainability of health systems [50] and improve the diagnostic process for patients [51]. However, this can only be achieved through the proper design and implementation of these services, which implies rigorous evaluations of health systems and of the impact of digitization on the beneficiaries of the systems [52].

## 2. Materials and Methods

The paper is based on two methods. The first research is qualitative and is related to the opinions, attitudes, and beliefs of Romanian experts on the use of eHealth and mHealth for the prevention, detection, and treatment of mild mental disorders. The second one quantifies the opinions, attitudes, and behaviours of Romanian adults on their openness to adopt new technologies for mental health management. 

The objectives for qualitative research were: (O_1_) to find out experts’ opinions on eHealth and mHealth; (O_2_) to identify opinions on the integration of “digitized mental health” in current health services; (O_3_) to determine opinions on the risks and benefits of implementing digitized mental health services in the detection and treatment of depression in its early stages; and (O_4_) to establish the profile of the consumer of health services in Romania. 

Based on the results obtained, the objectives and hypotheses of the quantitative research were established, to quantify the opinions, attitudes, and behaviours of Romanian adults regarding the use of digital tools and their openness to adopting new technologies for mental health management. Thus, the following objectives was formulated for quantitative research: (O_1_) to know the extent of using digital tools for health purposes by adults; (O_2_) to determine the extent of using the technology by adults for various purposes related to mental health; and (O_3_) to identify future consumption intentions of adults regarding digital tools for mental health purposes.

### 2.1. Qualitative Research

The method chosen is an exploratory survey among Romanian experts and was performed by interviewing competent, highly qualified people who have extensive experience in the field of the topic under investigation. The topic addressed the opinions and attitudes about eHealth and mHealth and their use in the prevention and treatment of mental illnesses, particularly in prevention and screening campaigns for early-stage depression. The research took the form of in-depth interviews based on open-ended questions to experts in the field: psychiatric doctors, family doctors, psychologists, and marketing managers of clinics/medical practices in Romania. 

The reason for choosing to conduct this type of research is the need to identify, in advance, the opinions of the experts in the field [53]. The methodology was based on interviewing for 60 min, based on the interview guide, using an in-depth interview method, 10 specialists working in the health field during the period of May–June 2021. The topic addressed the opinions and attitudes about eHealth and mHealth and their use in the prevention and treatment of mental illnesses, particularly in prevention and screening campaigns for early-stage depression.

Therefore, given the fact that, in the mental health area a beneficiary comes in contact with specialists of several types, the authors performed an exploratory survey among these specialists: family doctors (as they are the first experts with whom the patient comes in contact for potential medical problems); psychiatrists (they link the patient with a diagnosis); psychologists (experts in assessing the thinking, affects, and behaviours linking the patient to mild depression); healthcare marketing and PR personnel (experts who know both the consumer and the provider—hospitals, clinics, etc.).

These 10 specialists were online recruited (via platforms, such as LinkedIn and invitation e-mails sent to the hospitals/clinics headquarters), given the still existing social distancing restrictions due to COVID-19 pandemic in the data collection period (May–June 2021). Over 40 such invitations for participation were sent to specialists; however, the final positive response rate was of only 10 experts due to the complicated social and medical context existing at that time because of the COVID-19 pandemic. The sample for the qualitative research was restrained to only 10 specialists as the authors faced limitations in terms of the existing social and medical context, together with time, financial, and infrastructure limitations. 

To obtain detailed information about the consumer of health services and their behaviour, about the Romanian healthcare system, the implementation of eHealth and mHealth in the field of mental health in Romania, and the detection, monitoring, and treatment of mild depressive symptoms with the help of technology, and to have a better overview of the research theme, four categories of specialists were included in the sample:GP—the first experts with whom the patient comes in contact for routine or through medical check-up regarding potential medical problems. They are responsible for guiding the patient to offices/units/specialized doctors.Doctors specialized in psychiatry and mental and behavioural disorders—they make the direct link between the patient and the diagnosis, with a key role in medical processes.Psychologists—experts in assessing the thinking, affects, and behaviour that most often link the patient to mild depression.Personnel from the PR or marketing departments on the health market—experts who know both the consumer and the provider’s capacity (hospital, office and clinic).

The research aimed to determine detailed information about the consumer of health services and their behaviour, about the Romanian healthcare system, the implementation of eHealth and mHealth in the field of mental health in Romania, and the detection, monitoring, and treatment of mild depressive symptoms with the help of technology.

### 2.2. Quantitative Research

To conduct quantitative marketing research, the survey was chosen, and data was collected through a questionnaire. It was administered online using the Computer Assisted Web Interviewing (CAWI) method [54] and the sampling method was a non-random, rational sampling method combined with the snowball method. Data collection took three months, with the data collection period being from October 2021 to January 2022. The data collection was based on a questionnaire that included 29 questions, 23 of which were questions about the opinions, attitudes, and behaviours of Romanian adults regarding the use of digital tools to prevent depression, and six questions to identify the respondents. The sample was represented by a total of 514 subjects—women and men over 18 years old in Romania were interviewed.

## 3. Results

In the first section, the results of the qualitative research will be reviewed, followed in the second section by the results of the quantitative research.

### 3.1. Qualitative Research

Research has shown that 80% of the subjects consider that digitization in healthcare is a necessity, while 20% believe that digitization in the field comes with both pluses and minuses (“*It is a necessary good or bad, it depends on how much will be digitized in healthcare (…)*”). The majority of survey respondents believe that digitization in health (eHealth and mHealth) is important, particularly in the context of global dynamics: “*the population—in size and education—has a tremendous dynamic, and we need tools to facilitate its access to health services*”. Subjects consider that, without technology, medical research would not evolve, communication would be hampered, medical data transfer would raise issues, etc. 

The evolution and performance of health systems are ideals that depend on technology and the digital context. In the opinion of the subjects, the majority consider eHealth and mHealth as necessary tools in the evolution and performance of health systems in the current context. Subjects consider that Romania is at an early stage in terms of digitization in healthcare, particularly in comparison with the healthcare systems of other EU Member States. There are timid attempts from this point of view, there are also failed attempts; however, what is important is the desire to take the step towards digitization “(…) there is openness and steps are being taken in this direction, although in a chaotic and uncoordinated way”). 

Some subjects consider that our country’s incipient stage of digitization in health is caused by the lack of investment in infrastructure, in IT networks, as well as the lack of education and digital skills of the population. Some of the respondents believe that Romania needs time and massive investments for the healthcare system to be able to be compared with Western countries in terms of digitization: “It is true, there are attempts to digitize the system, we have the capacity to store and transmit medical data between clinics, hospitals, we also have the health card system, as it works, better or worse, but if you ask me about digitized diagnoses and treatments, I will ask you to repeat the question in a few years”. The majority of the subjects of this research consider that, in Romania, the stage of development of actions for the prevention, detection, and treatment of depression is at an early stage, particularly in terms of public actions. 

They consider that the spectrum of mental disorders is not yet given due importance. They mention that, in Romania actions to detect and treat depression at an early stage have remained at the paper stage. At the same time, the subjects also mention the problems related to the lack of trained family doctors, the lack of clear statistics in the field, the lack of vision in the construction of the national mental health programme, too few psychiatric centres, etc. Moreover, all the research subjects underline the lack of effective prevention campaigns in this field performed by the public health system in Romania. 

Subjects consider problematic the level of knowledge of methods of detection and monitoring of mental and behavioural disorders among Romanians and differentiate the population by area of residence. They consider that the problems are even greater in rural areas, due to the stigma associated with this field, financial problems, lack of resources for information, etc. Some of the subjects consider that the Romanian population, particularly in urban areas, knows the risk factors for these disorders; however, they justify the lack of action in this regard with arguments, such as stigma associated with mental disorders, ignorance, and fear of doctors, etc. All subjects consider that, in Romania, the lack of health education is a key problem that is reflected in the low level of knowledge and awareness of methods of prevention, detection, and monitoring of mental disorders of Romanians. 

At the same time, the subjects explain Romania’s early stage in this regard with the lack of coherent, sustained, and effective campaigns in this field. When asked what eHealth or mHealth technologies are used in the medical field (particularly in the area of mental and behavioural disorders) in Romania, at the moment, the subjects did not mention other specific eHealth or mHealth technologies apart from those implemented in other medical fields. 

Thus, all subjects mentioned the National Health Card System and the electronic patient record. A total of 40% of the subjects also mention e-prescribing systems, four subjects also referred to private medical units’ systems for scheduling and reporting medical results, while only two mentioned communication applications and one subject mentions data transfer and storage systems, as well as applications for big data and research.

Eight out of ten research subjects consider that eHealth and mHealth can contribute to the quality and efficiency of detection and treatment of mental and behavioural disorders, while two subjects believe that these technologies contribute to the quality and efficiency of current health services only through the role that technologies have acquired in communication and information. 

A part of the subjects surveyed believes that technologies already contribute to the quality and efficiency of health care in mental disorders, generating benefits both economically and in optimising the information flow of patient–doctor, doctor–doctor, and doctor–patient. At the same time, the subjects consider that eHealth and mHealth can bring essential contributions to the quality and effectiveness of mental and behavioural disorders prevention campaigns.

From the point of view of the advantages and risks of technology-assisted care in the sphere of mild mental and behavioural disorders as opposed to face-to-face interactions, mainly the advantages mentioned by the subjects refer to the influence that technology-assisted care can have in reducing the stigma associated with these disorders, improving the accessibility of the population to health services, cost reductions, and the positive impact generated by rapid communication, data transfer, and storage. 

On the other hand, most subjects see data security and control as risks in this type of care; however, more than that, major risks are also the difficulties in personalizing the medical service, the impossibility of establishing a doctor–patient relationship based on empathy, as well as limitations related to education or functional illiteracy among the population. Most subjects believe that eliminating face-to-face human interaction would eliminate fears and reduce the stigma associated with depression. Several respondents believe that the impact would be positive only if there is a well-developed system, managed by professionals in the field, combined with improved education and a more responsible attitude of the population in this area. 

More than half of the respondents to this question believe that eliminating face-to-face human interaction is not appropriate, as technologies can neither replace doctors nor interpret non-verbal language or provide empathy. We found that 30% of the subjects take a neutral view, stating that eliminating human interaction for the prevention, detection, or treatment of mild depression has both advantages and disadvantages (“(…) it may help, but it will not be able to replace the doctor–patient relationship, even when we are talking about less serious forms of some conditions”).

The research subjects consider that the evaluation of web or mobile applications for mental health should be assessed by interdisciplinary teams of specialists from the Ministry of Public Health, specialists from international bodies, IT specialists, and doctors (family doctors and psychiatrists) and psychologists, based on criteria related to effectiveness, safety, usefulness, accessibility, potential therapeutic effect, and data/information quality. In their view, endorsement should comply with national and international standards, both medical, ethical, and informational. For all research subjects, the issue of data security and privacy is an important one in the evaluation and endorsement process of this type of applications. Some of the subjects consider, however, that assessing the usefulness and interoperability of these applications is difficult, because “(…) usefulness is difficult to measure before launching applications on the market, and the medical field is a sensitive one.”

Half of the subjects find the implementation and combination of “digital mental health” with current clinical care difficult to achieve, particularly in family medicine, mostly in rural areas. The majority see timely and easier implementation in the area of data transfer, data storage, and communication, considering that there is a need for profound changes in the system and societal mindset, for digital healthcare to truly complement current clinical care (“We need to rebuild the whole system and reorganize all the health laws so that there is a digital—clinical care combination in a way that is valid, safe and generates positive effects.”). 

Subjects find digital health services useful not only in communication and data transfer but also in the process of monitoring patients suffering from mild forms of mental or behavioural disorders. The implementation and combination of digital healthcare services with current clinical care is desirable and could be achievable with sustained efforts in the future in Romania as well, in the view of the research subjects.

### 3.2. Quantitative Research

The quantitative research showed that 25.5% of all respondents (N = 514) were generally not interested in health issues. The majority, however, representing 74.5%, confirmed their general interest in health issues.

The determinant reason for the last visit to the doctor represented graphically in Figure 2 illustrates the situation of the 35.6% of respondents (183 out of 514 sample members) who went to the doctor for routine check-ups, while the remaining 64.4% (331 respondents) visited the doctor due to health problems. Regarding the experience of their last visit to the doctor, respondents were asked to describe it in a single word, using the word association technique. Thus, a large proportion of respondents described the experience using words referring to financial aspects, such as “money”, “expenses”, “costs”, “bribe”, etc., while others referred to physical (“pain”, “fractures”, “operations”, “accident”, etc.) or emotional aspects (“fear”, “fright”, “stress”, “judgement”, etc.).

Another relevant result is the statistic on the means used by respondents to inform themselves about health issues in general. For this purpose, a question constructed on a nominal scale with multiple choice was used, and the results are shown in Figure 3.

TV (15.9%), the Internet (15.8%), and health professionals (14.2%) make up the top three most-used sources for health information in general by participants in this research. This is followed by family (14%), friends (13.4%), flyers (7.8%), and mobile apps and books or magazines (6.4%). Radio (6.1%) is the last source of information used, and one respondent indicated “colleagues” as a source of information for general health issues.

In terms of mental health, interest in mental health issues was determined by a question constructed on a binary scale, the analysis of the primary data indicated that 61.28% of the 514 participants in this research were concerned about mental health issues, while the remaining 38.72% said they were not concerned about mental health issues. We also sought to determine the degree of importance attached to mental health by adults. For this, a question was constructed using an interval scale with bipolar adjectives of opposite meanings, where level 1 indicates the answer “Very little” and level 5—“Very much”.

Of the 341 valid responses received from sample members, 6.16% attach very little importance to their mental health; 24.63% of respondents give a great deal of importance to their mental health, while 17.30% of respondents say they give very much importance to their mental health. At the same time, at the sample level, 23.75% of respondents attach little importance to this aspect. Most respondents to this question (28.2%) said that they care neither greatly nor little about their mental health. The average value of the respondent ratings of the degree of importance attached to their mental health is 3.23 points on a scale of 1 to 5; the median and the module are 3, while the standard deviation is 1.171.

Another relevant result of the survey is the statistic on the means used by respondents to inform about their mental health. To determine the most used means of information for this purpose, a question constructed on a multiple-choice rating scale was used and the results are shown in Figure 4.

One of the statistical hypotheses of the present research assumed that there is no concentration of responses regarding the sources used for mental health information. (H0: O_*i*_ = E and H1: O_*i*_ ≠ E_*i*_.)

**H0:** 
*There is no concentration of responses on sources used for mental health information.*


**H1:** 
*There is a concentration of responses on sources used for mental health information.*


Table 1 shows the differences between the observed and expected frequencies of the response variants. Following the application of the χ^2^ test for hypothesis testing in the case of univariate analysis, it can be guaranteed with a 95% probability that the distribution of responses of the population under investigation differs from the uniform distribution. There is a concentration of responses on the response variants “Medical staff”, “Family” and “Friends” and a lower concentration on the response variants “Books and journals”, “Flyers”, etc. χ^2^ =379.463 a is higher than χ^2^ = 14.067 (calculated with the CHIINV function in Excel), which leads to the acceptance of the alternative hypothesis H_1_, which means that the frequency distribution is different from uniform, i.e., there is a concentration of responses on sources used for mental health information. The same decision can also be made based on Asymp. Sig = 0, which is smaller than α = 0.05 (Table 2).

Another question of the questionnaire aimed to obtain information on the use of digital tools, such as mobile apps, websites, electronic devices, etc. for health-related needs (information, physical activity monitoring, doctor appointments, etc.) (Figure 5).

To obtain information on the consumption behaviour of digital tools for different health-related needs, respondents were asked to indicate the frequency of consumption (1 = “Very rarely or not at all” and 5 = “Very frequently”) of these tools for monitoring the effort and improving physical activity; monitoring health problems; receiving medication notifications; measuring, recording or transmitting data about medication and/or treatment; informing about symptoms, causes and treatment methods; and communicating with the doctor or others about medical topics (Figure 6).

Of the 514 respondents, the majority (32.30%) said that, in the past year they have used digital tools very rarely or not at all to monitor and improve their physical activity. At the other end of the scale, those who said that they had used such tools very often in the last 12 months to monitor and improve their physical effort represented 27.24% of the total sample.

Indicators of descriptive statistics show an average score of 2.90 points on a five-point scale (1 meaning ‘Very rarely/not at all’ and 5, ‘Very often’) for the use of digital tools for monitoring physical activity and improving physical activity in the last 12 months. Of the total sample members (514), 36.77% used digital tools very rarely or not at all to monitor health problems. At the other end of the scale, only 9.92% of all respondents said that they had used digital tools very often in the last year to monitor health problems (Figure 7).

At the sample level, 20.62% said that, in the past 12 months, they occasionally used digital tools to monitor health-related issues, on par with those who rarely used digital tools for this purpose in the past year, while almost half of the sample (47.28%) said that, in the past 12 months they rarely or not at all used digital tools to receive medication notifications. Only 7.39% of all research participants said that, in the past year they had used digital tools very often to receive medication notifications (Figure 8).

Of the 514 respondents, the majority (27.82%) stated that, in the last year they used digital tools very often for information about symptoms, causes, and treatment methods (Figure 9). On the other hand, those who said that they had used such tools very rarely in the last 12 months for information about symptoms, causes, and treatment methods represent 15.95% of the total sample.

Of the total sample members (514), 25.49% rarely used digital tools to communicate on medical topics. A total of 21.98% of all research participants stated that they rarely or not at all used digital tools for this purpose in the last year (Figure 10).

As can be seen in Figure 11, data security (19.9%), content (19.6%) and accessibility (16.3%) are the three most important determinants of the use of digital health information tools nominated by the participants in this research.

Another relevant result of the research refers to the intention to use the technology and its tools for the health-related needs of the adult inhabitants in Romania (Figure 12).

By applying the t-Student test in the case of univariate analysis, t_calc_ = 10.727 was obtained, which does not belong to [−1.96; +1.96], and thus it requires the acceptance of the alternative hypothesis, H_1_. This decision is also confirmed by comparing the level of significance of Sig. (2-tailed) with the theoretically considered significance level of 0.05. Since the level of significance Sig. (2-tailed) is less than the theoretical significance level (0 < 0.05), it can be guaranteed, for a probability of 95%, that the percentage of the adult population that intends to use technology tools for health needs is different from 50%. 

Deepening the issue of future behaviour in terms of technology for health needs by using a question constructed on an interval scale with opposite bipolar adjectives (1 = “Very unlikely” and 5 = “Very likely”) the respondents were likely to use the technology for different purposes. At the sample level, 34.34% of respondents stated that they are very likely to use technology tools in the future to monitor effort and improve physical activity. At the same time, 26.46% of them stated that the use of technology for the above reason is very unlikely. 

A total of 16.54% of all research participants indicated that they are unlikely to use technology to monitor and improve physical activity in the future, while 8.56% are likely to use technology for this purpose. We found that 14.2% of all respondents indicated the third level of the scale signifying that they are undecided on this issue. The descriptive statistical indicators show an average of the intention to use the technology tools for monitoring the effort and improving the physical activity of 3.08 points on the mentioned scale. The median is equal to 3, and the mode is given by the value 5. The standard deviation is equal to 1.64, and the variance = 2.68.

Another interesting result is that most respondents (30.54%) do not know if they will use the tools of technology in the future to monitor mental health issues while less than 10% of them identified with the answer “most likely”. As can be seen in Figure 13 of the total respondents, the majority (26.65%) are undecided on the future use of the technology to receive alerts or reminders about the follow-up of treatment. We found that 25.10% stated that they are very unlikely to use the technology in the future for this purpose, and 26.07% stated that they are unlikely to use the technology for alerts and reminders to follow the treatment.

Given the results obtained thus far in this research, an interesting outcome is illustrated by Figure 14, which shows that the majority (27.04%) say they will probably use technology to learn about aspects of depression, and 25.29% of all respondents say they are likely to use the technology for this purpose in the future.

Regarding the openness of the sample members about the intention to use the technology for measuring, storing, and transmitting medication data, most of the respondents, i.e., 25.10%, are undecided. The result is of particular interest because (Figure 15) there was a similar percentage of the respondents who identified with the very unlikely answer (20.04%) and respondents who gave the most likely answer (18.29%).

Regarding the use of technology in the future for access, storage, and transfer of personal health data to a doctor, 25.88% of respondents are undecided. Moreover, the percentage of 16.34% who chose the very unlikely answer option cannot be unnoticed (Figure 16).

The relationships between determinants (characteristics) in the use of mobile applications for mental health and the intention of adults to use technology tools for health-related needs were also highlighted. The statistical hypotheses are

**H0:** 
*There is no link between the determinants of adult use of technology and the intent to use technology for health-related needs.*


**H1:** 
*There is a link between the determinants of adult use of technology and the intent to use technology for health-related needs.*


For testing, the application of the binary logistic regression method was chosen in this sense, as this type of regression can illustrate realities between future specific actions and current behaviours. Table 3 illustrates the factors that would lead respondents to use technology to learn about mental health. These are the independent variables in the model (X), while the intention to use the technology for mental health needs is the dependent variable (Y). The dependent variable is constructed on a binary scale (1 = “Yes”, 0 = “No”), and the other variables were recoded, moving from the nominal multiple-response scale to the binary scale.

From the 514 members of the sample, 1478 responses were obtained. Data security (19.9%), content (19.6%), and accessibility (16.3%) are the most nominated features. The least nominated were the characteristics related to the location functions of medical centres and specialized offices (7.6%) and other characteristics (0.1%). From Table 4, 71.4% of respondents intend to use the technology for health-related needs in the future. Attempts will be made to determine the influence of the aforementioned factors on this intention using the logistic binary regression model of the SPSS system. The model parameters (α and β) are presented in Table 5.

Based on the results in Table 5, the model equation was constructed:(1)$$P\;\left(Y=\frac{1}{x_{1}}\right),\;x_{2},\;x_{8}=$$
(2)$$\frac{e^{-2.249+1.405Acc+0.663Sd+2.008Ct+1.778Co+3.096Mf+0.281Fl+2.066Sync-18,954Alt}}{1+e^{-2.249+1.405Acc+0.663Sd+2.008Ct+1.778Co+3.096Mf+0.281Fl+2.066Sync-18,954Alt}}$$

The statistical significance of the model parameters is observed in the same table. Since, for the parameters Alt and Fl, the significance level Sig. > 0.05, it cannot be confirmed, for 95% probability, that these two variables of the model are statistically significant.

The same table also contains relevant information on the real influence of independent variables on the likelihood of using technology for health-related needs (column Exp (B)). The values of Exp (B) are relevant because they illustrate how often the probability of the event occurring increases when the predictor increases its value by one unit. Relevant is also the change ratio in the sample, (Exp (B) − 1), for a detailed explanation of the influences.

Thus, accessibility can increase, by more than 300%, the chances that an adult will use the technology for health-related needs because of the influence exerted by this characteristic on them (4.074 − 1 = 3.074). Data security is a feature that increases the chances for an adult from Romania to use technology for health-related needs by 94.1% (1.941 − 1 = 0.941).

Content is the most influential feature in increasing the chances of an adult using technology for health-related needs (7.450 − 1 = 6.450). To be a significant parameter, it must have a very low level of significance (Sig. < 0.05) and a Wald value greater than 0.5. Thus, the analysis shows that the location functions of medical centres and clinics (Fl), as well as other characteristics (Alt) are not statistically significant; therefore, they cannot be considered important in the present analysis, and they can be excluded.

The significance of the model can be described by the value of R-Square. This indicates the ratio between the influence of regressors on the total variance of the dependent variable. The R-square value must be as close as possible to 1. In Table 6, the values of the correlation coefficient are calculated by two methods, Cox & Snell, and Nagelkerke R Square.

For the Cox & Snell method, the value of the correlation coefficient is 0.408 (40.8%), while by the Nagelkerke calculation method, R Square has a value of 0.585 (58.5%). This means that the built model has average explanatory power. However, the model can be improved by considering other variables as well. As a result, it is advisable to reconstruct the model by considering other variables to increase the explanatory power of the model.

Although the power of the model is average, about half of the total variance of the dependent variable is explained by the proposed model of health. A hypothesis was also tested to determine the existence/non-existence of a link between the age of adults and the extent to which they agree with a statement regarding the use of digital tools to prevent depression.

**H0:** 
*There is no link between age and level of agreement with the statement “The use of digital tools helps prevent depression.”*


**H1:** 
*There is a link between age and level of agreement with the statement “Using digital tools helps prevent depression.”*


To test the hypothesis, the Kolmogorov–Smirnov test was applied in the case of the two-way analysis. Thus, the differences between the observed and the expected distributions, cumulatively increased, were tested. For the present analysis, the sample was grouped into two categories: a group of persons under 49 years of age and a group of persons over 49 years of age (Table 7).

Differences in frequency distribution can be observed between the two groups. To determine the statistical significance, the Kolmogorov–Smirnov test was applied based on the following hypotheses:

**H0:** 
*The maximum difference between the cumulative frequencies for adults under 49 years (F1) and those over 49 years (F2) is zero.*


**H1:** 
*The maximum difference between the cumulative frequencies for adults under 49 years (F1) and those over 49 years (F2) is non-zero.*


Since the value D = 13.31% is lower than the value D_calc_ (50.5%), the null hypothesis will be rejected, which means that the maximum difference between the cumulative frequencies for adults under 49 years (F1) and over 49 (F2) is non-zero. The same conclusion can be drawn from the Asymp. analysis. Sig. 92-tailed (0.000) has a value below the significance level of 0.05, which means accepting the alternative hypothesis. Thus, there are differences in age groups regarding the level of agreement expressed with the statement “The use of digital tools helps prevent depression”.

## 4. Discussion

Recalling the objective presented in the first part of the methodology, we conclude this section by introducing the big picture of the study. They placed together the statements for achieving and the results obtained for qualitative research method (Figure 17) and quantitative research method (Figure 18).

The scientific value of the research consists in aligning the continuous evolution of technology in the health services market [2,10] with the need of presenting the main issues of public health environment in Romania [4,5] to promote opportunities regarding eHealth and mHealth in campaigns to prevent and raise the awareness of mild mental disorders based on the consumer profile of health services as a part of an integrated health management system [6,7].

Qualitative and quantitative research was performed wherewith the authors discovered and quantified the opinions, attitudes, and behaviours of adults on the use of digital tools and openness to the adoption of new technologies for managing mental health. The study enriches the scientific literature by adding a new perspective to the current literature [14,15,21]. First, most of the hypotheses from which the qualitative research started were confirmed.

The interviewed experts mentioned the stigma associated with mental illness, as well as the lack of resources (financial and time) among the main reasons why Romanians ignore their concern for their mental health. eHealth and mHealth services are useful [10,11]; however, the Romanian medical system is still deficient in terms of full integration of the two concepts. The interviewed experts have a generally positive attitude towards the two concepts; however, they believe that the Romanian medical system needs a consistent improvement of the infrastructure, particularly in education so that the two concepts can be integrated and exploited to their maximum potential in the country.

The eHealth and mHealth services impose a technical and personnel infrastructure [8], that the Romanian medical system does not yet have, although efforts are being made in this direction. This hypothesis is partially confirmed, as there is an opinion at the sample level that there are IT performers and capabilities in terms of ICT infrastructure; however, there is a lack of investment in medical infrastructure and human resources in the field of health, to create functional systems and digitized mental health now.

Regarding the behaviour of Romanian consumers of health services, they avoid specialized assessments for their mental health due to the stigma associated with this specialization, fear, or shame of being judged or categorized as “crazy” is the main reason for their lack of action in this regard. 

These reasons are often compounded by the lack of accessibility to such services, either due to lack of information (due to lack of campaigns in the field and health education) or lack of infrastructure: few medical centres, distance from medical facilities, lack of temporal and financial resources. Experts believe that beyond the improvement of the factors listed above, the family and the entourage are the ones who have a major influence on consumer behaviour.

At the same time, to determine the Romanian consumer to resort to specialized services for the evaluation of mental health, it is necessary to have constant information campaigns [16], as well as to improve the health education at the level of the population [15]. Technology tools can have major advantages in achieving these two goals.

At the level of the selected sample, the general opinion is that Romanians’ interest in preventing, detecting, and treating depression will evolve in a positive way, not only because of the increasing importance given to the subject globally but also because Romanians become more involved in facilitating the access to information, and technology is gaining more meaning for this purpose, an example being the increase in the number of users of lifestyle applications and physical exercise monitoring in Romania.

Even so, increasing the interest of Romanians and undertaking efforts to prevent, detect and treat depression require support from the state, medical units, and NGOs by conducting effective campaigns in the field and providing the necessary infrastructure and authority [39,40] to achieve these goals.

The analysis of the primary data from the quantitative marketing research undertaken revealed valuable information for the development of marketing policies and strategies specific to the field. Thus, although most of the participants in the research stated their general interest in health issues, more than a quarter of the sample members (25.5%) did not pay attention to health issues.

Only 35.6% of research participants go to the doctor for routine assessments, most visit doctors only because of health problems and 27.3% of adults say they are rather dissatisfied with their general health. Regarding the sources used for health information, TV (15.9%), the Internet (15.8%) and medical staff (14.2%) are the most used means of information according to the analysis at the sample level. Surprisingly, when it comes to mental health, the research found that the Internet (20.5%), friends (18.2%) and family (13.1%) are the most widely used means of information used.

An important aspect illustrated by the analysis of primary data is related to the importance given to mental health: 23.75% of research participants give little and 6.16% of participants very little, importance to their mental health. Almost a third of respondents believe that the use of applications and gadgets to monitor effort, stress, diet, sleep, etc. means a concern for mental health while 67.9% of research participants have never participated in a special assessment for their mental health.

The analysis of primary data has also generated valuable information on digital health tools. More than half of the respondents said they use digital tools for health needs. 25.68% of participants believe that technology is very important for informing about mental illnesses, such as depression, while 27.04% consider technology important for this purpose.

One-third of respondents, however, used digital tools very rarely or not at all to monitor and improve their physical activity. 36.77% used digital instruments very rarely or not at all to monitor health problems.

On the other hand, only 9.92% of all research participants stated that, in the last year they have often used digital tools to monitor health problems. Only 7.39% of all research participants stated that, in the last year they have often used digital tools to receive notifications for medication. 38.91% stated that, in the last 12 months they used digital instruments very rarely or not at all for measuring, recording, or transmitting medication/treatment data, and 27.82% of all respondents used digital instruments very often for training on symptoms, causes, methods of treatment.

At the same time, data security (19.9%), content (19.6%) and accessibility (16.3%) are the three most important characteristics of the use of digital tools for mental health information nominated by participants in this research. These facts bold the situation found in the scientific literature [26,28,37]. 71.4% of the sample members intend to use the technology and its tools for health needs in the future; however, most of the respondents do not know if they will use the technology tools to monitor mental health issues.

Although a non-random sampling method was used and the results could not be extrapolated to the total population, it was considered important to prove, in the sample, links between population characteristics and certain aspects of the research topic, through analyses of inferential statistics.

Thus, statistical tests concluded that 65% of the researched population is concerned with mental health; the percentage of the adult population in Romania that knows about prevention campaigns against depression is less than 40%; over 50% of the adult population intends to use technology tools for health-related needs.

At the same time, according to the results of the research, there is a focus on “medical staff”, “family” and “friends” and a lesser focus on the answer “books and magazines” and “flyers”. There is a link between the age of adults and the extent to which they agree that mental health care is manifested using technology tools (applications and gadgets for monitoring effort, stress, sleep, nutrition, etc.), in the sense that younger people tend to be more positive about it than older people.

Technology and health education are key elements in promoting methods of preventing and combating mental health disorders; however, the personalization of the message, which can be best performed by doctors, must be considered. In addition, accessibility, data security and quality content are factors that influence the well-being of both young and older adults.

However, the challenges of implementing digital tools as part of the integrated health management system can be financial, educational (on health and on ITC), and the lack of programs in this direction. On the other hand, the challenges in the adoption of eHealth and mHealth are the lack of specialists and infrastructure, lack of patient-centred services; accessibility (rural areas); and education.

Another critical aspect that influences the mental health management system from the point of view of the use of digital technologies is the digital divide. The challenges that people, particularly the elderly with a negative perception about their age are not favourable factors for the use of eHealth and mHealth technologies. Moreover, policies to promote a system in which mental health is managed through these digital tools would have no effect on people who refuse to adopt technology not only for their own health care but also for other daily activities.

These are the elements and real challenges that mental health management system must consider in eHealth and mHealth adoption. Therefore, the authors promote a menta health management improvement with marketing strategies to deploy policies and strategies to combat the aforementioned challenges.

## 5. Conclusions

The scientific novelty is represented by the identification of the opinions of some experts regarding the concepts of eHealth and mHealth but also by their opinions regarding the integration in the current health services, particularly in the sphere of mental and behavioural disorders. The research also sought to determine ways to stimulate mental health care and identify opinions on the risks and benefits of implementing digital health services in the prevention, detection, or treatment of depression at an early stage.

The academic community can contribute by promoting the results of research conducted in the field for the responsibility of institutions and consumers on mental health, as well as raising the awareness of the importance of technological progress in the development of sustainable health programs and systems when combined with social involvement.

Managerial implications target both macro (state and ministry) and micro decision-makers (hospitals, GPs, psychologists, and psychiatrists), which are the entities that can influence the development of the integrated management system for the emergence of opportunities in terms of the business environment, telemedicine, connection between state, the medical sector, and the manufacturers of equipment for the development of the field. The opportunities for decision makers are related to the develop policies and programs in this field and open new markets, as the consumers tend to make the transition to digital, and they manifest openness to try.

Within the marketing research performed, limits were also encountered. The first limitation of the research is the impossibility of extrapolating the results due to the chosen sampling method and the data collection technique that the researchers used. It is important to note, however, that the results obtained are relevant to the chosen topic, and the sample size of this research is a favourable factor for substantiating marketing policies and strategies appropriate to the field.

The second limit is produced by the many closed-ended questions in the questionnaire, with the accuracy of the research influenced by the limitations of the possible answers. Other limitations of this research refer to the qualitative research conducted, which presents both the uncertainty of the applicant regarding the sincerity of the participants in the in-depth individual interviews and the increased degree of sensitivity of the chosen topic and questions to participants. Despite these limitations, the marketing research undertaken can provide a solid basis for future research relevant to the topic, and the results obtained can contribute to the foundation and proposal of policies, strategies, and actions relevant to the health field to improve health system management.

In the future, the authors encourage the development of a comprehensive health marketing program involving the collaboration of public health institutions, educational institutions, and authorities responsible for digitizing Romania to implement digitized health education. This would contribute to health management. We also suggest the development of a long-term integrated health marketing program at the national level involving tools of communication technology to promote mental health and prevent depression in Romania. Extensions of the qualitative research performed at the regional level and the comparative analysis of the results, to obtain a complete picture of the analysed phenomenon would capitalize even more on the results of this research.

## Figures and Tables

**Figure 1 ijerph-19-09172-f001:**
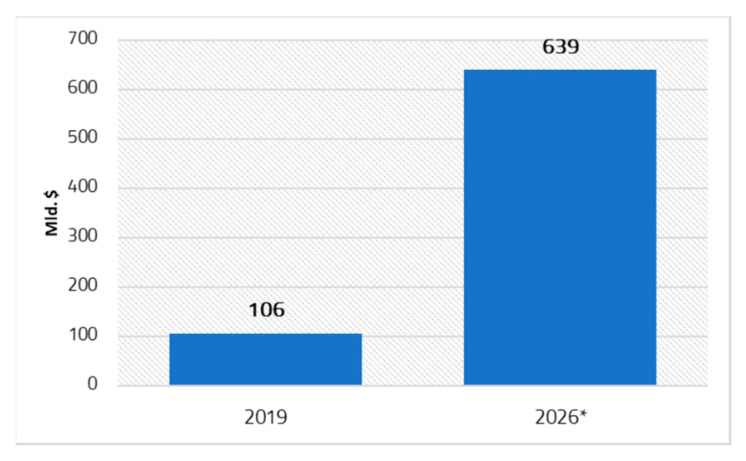
Global digital health services market in 2019 and forecast to 2026. Note: * forecasting.

**Figure 2 ijerph-19-09172-f002:**
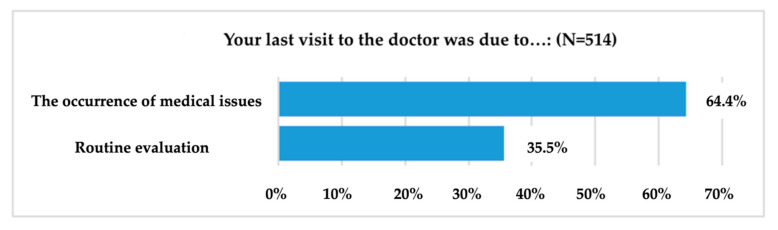
The determining reason for the last visit to the doctor.

**Figure 3 ijerph-19-09172-f003:**
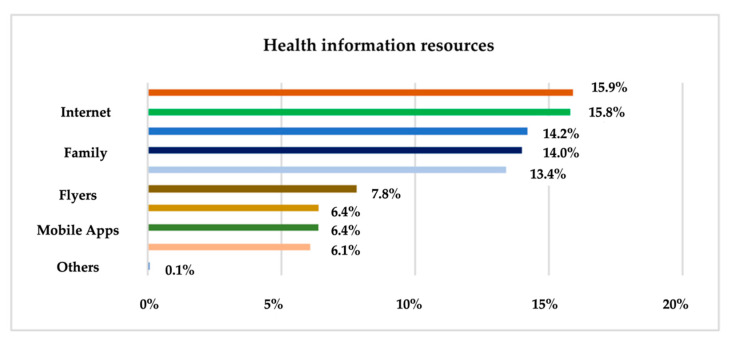
The sources used for health information.

**Figure 4 ijerph-19-09172-f004:**
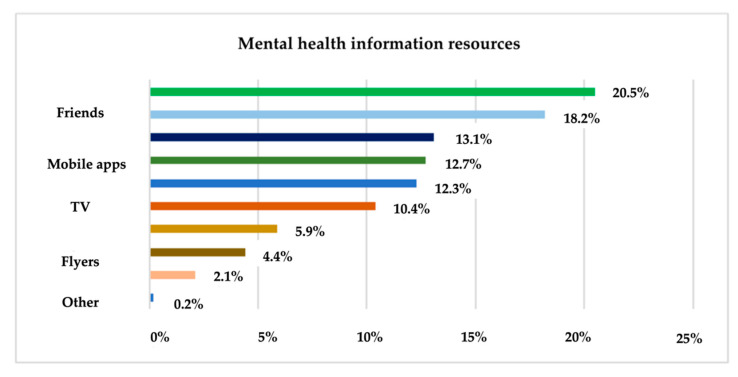
The sources used for mental health information.

**Figure 5 ijerph-19-09172-f005:**
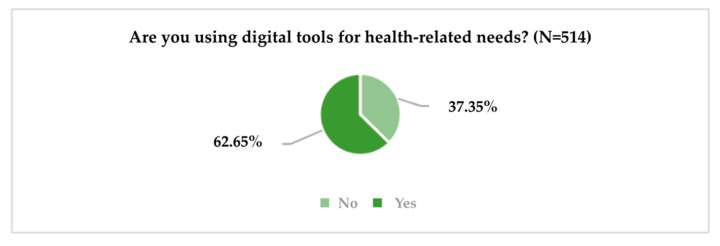
Use of digital tools for health needs.

**Figure 6 ijerph-19-09172-f006:**
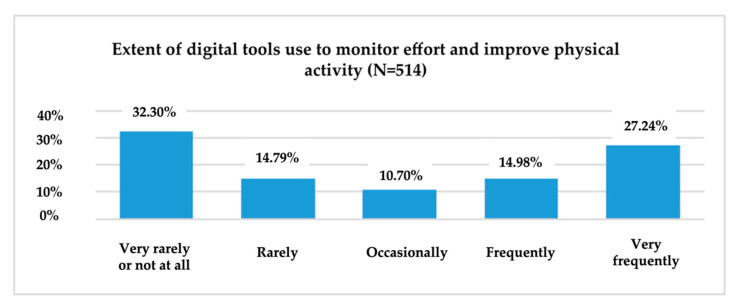
The use of digital tools to monitor and improve physical effort.

**Figure 7 ijerph-19-09172-f007:**
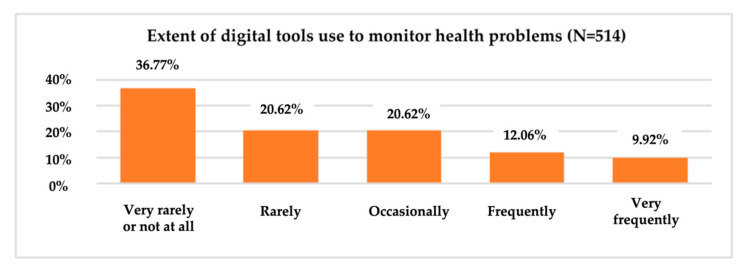
The use of digital tools to monitor health problems.

**Figure 8 ijerph-19-09172-f008:**
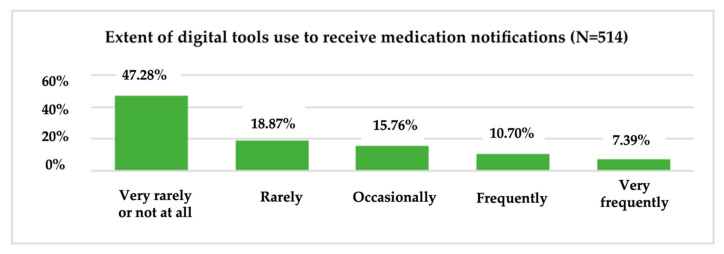
The use of digital tools to receive medication notifications.

**Figure 9 ijerph-19-09172-f009:**
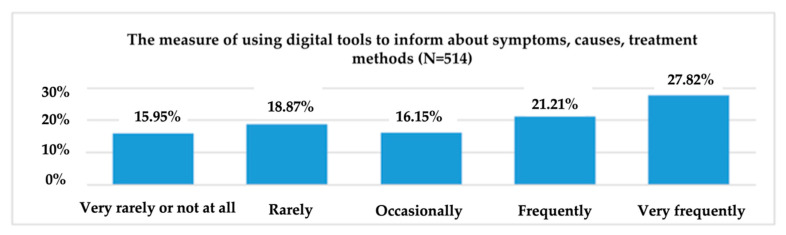
The use of digital tools for information about symptoms, causes, treatment methods.

**Figure 10 ijerph-19-09172-f010:**
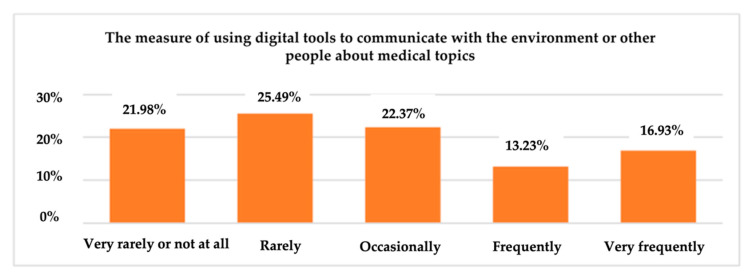
The use of digital tools for communication on medical topics.

**Figure 11 ijerph-19-09172-f011:**
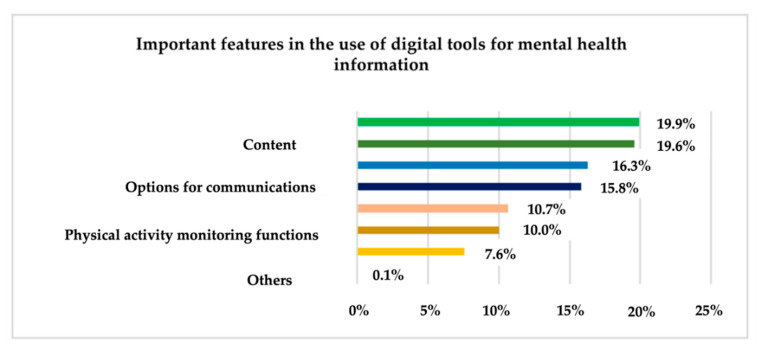
Features in the use of digital tools for mental health information.

**Figure 12 ijerph-19-09172-f012:**
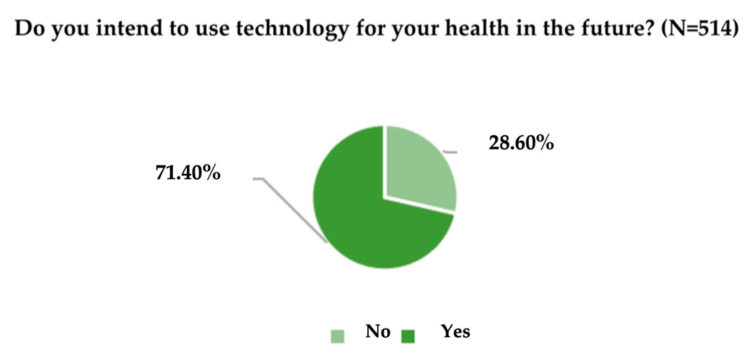
The intentions to use health technology.

**Figure 13 ijerph-19-09172-f013:**
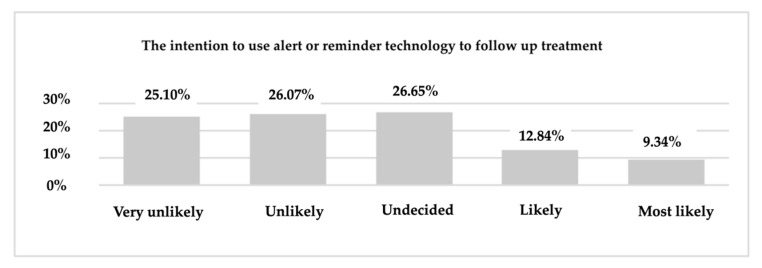
The intention of using technology—alerts or reminders for follow-up treatment.

**Figure 14 ijerph-19-09172-f014:**
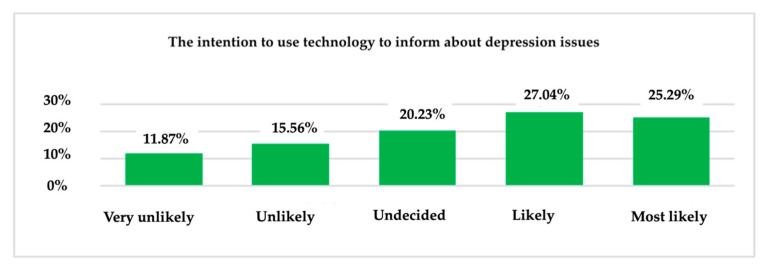
The intention of using technology—information about depression issues.

**Figure 15 ijerph-19-09172-f015:**
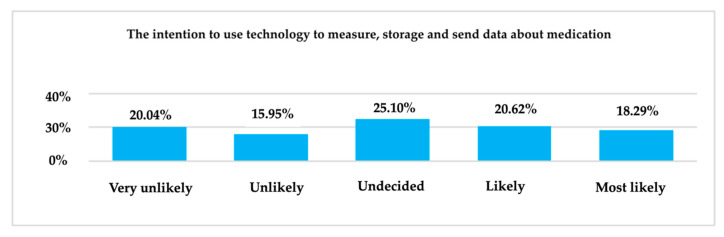
The intention of using technology—measurement, storage, and transmission of medication data.

**Figure 16 ijerph-19-09172-f016:**
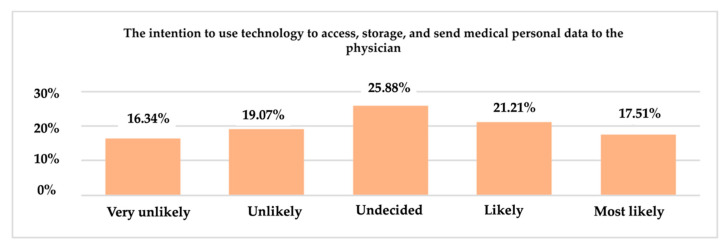
The intention of using technology—access, storage, and transfer of personal health data to a doctor.

**Figure 17 ijerph-19-09172-f017:**
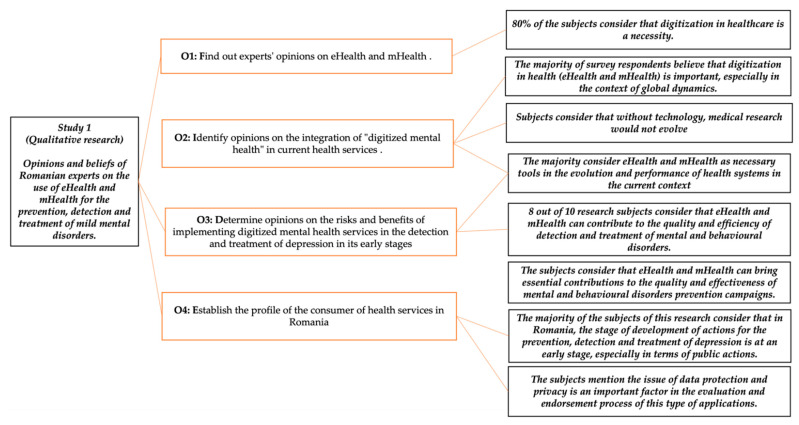
The first study (qualitative research overview).

**Figure 18 ijerph-19-09172-f018:**
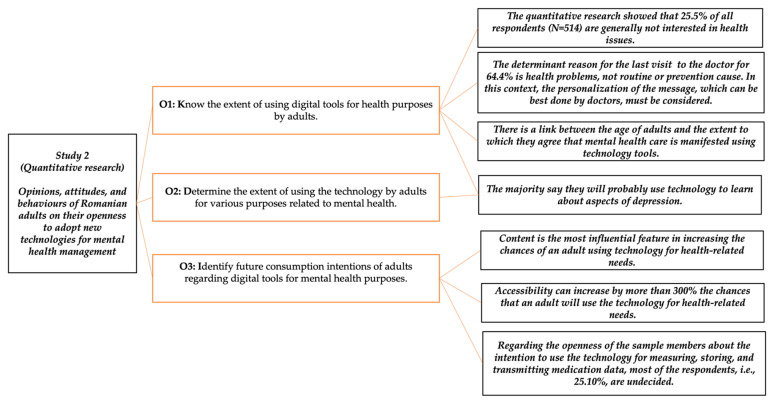
The second study (quantitative research overview).

**Table 1 ijerph-19-09172-t001:** Observed frequencies and expected frequencies (sources of information on mental health).

	Observed Frequencies N	Expected Frequencies N	Residual Values
Books and journals	2	42.1	−40.1
Flyers	8	42.1	−34.1
Internet	9	42.1	−33.1
Mobile applications	25	42.1	−17.1
Friends	75	42.1	32.9
Family	93	42.1	50.9
Healthcare professionals	123	42.1	80.9
Other resources	2	42.1	−40.1
Total	337		

**Table 2 ijerph-19-09172-t002:** Chi-Square test for testing hypothesis 6.

	Sources of Information on Mental Health
Value of the Hi Square test	379.463 ^a^
Degrees of freedom (df)	7
Significance level (Asymp. Sig.)	0.000

^a^ 0 cells (0.0%) have expected frequencies less than 5. The minimum value of expected frequencies is 42.1.

**Table 3 ijerph-19-09172-t003:** The importance of technology tool features.

		Responses		
		N	% of Participants	% of Total Cases
Factors (Features) technology use	Accessibility (Acc)	241	16.3%	46.9%
	Data security (Sd)	294	19.9%	57.2%
	Content (Ct)	290	19.6%	56.4%
	Communications options (Co)	233	15.8%	45.3%
	Physical activity monitoring functions (Mf)	148	10.0%	28.8%
	Location functions of medical centres/profile offices (Fl)	113	7.6%	22.0%
	Synchronize data with already used applications (Sync)	158	10.7%	30.7%
	Other (Alt)	1	0.1%	0.2%
Total		1478	100%	287.5%

**Table 4 ijerph-19-09172-t004:** The intention of using technology for health needs.

		Absolut Frequencies	Percentages (%)	Valid Percentages (%)	Cumulative Percentages
Valid	Yes	147	28.6	28.6	28.6
	No	367	71.4	71.4	100
	Total	514	100	100	-

**Table 5 ijerph-19-09172-t005:** Variables included in the model equation.

								95% Confidence Interval for Exp (B)	
		B	S.E.	Wald	Degrees of Freedom (df)	Level of Significance (Sig.)	Exp (B)	Lower	Upper
Step 1 ^a^	Acc	1.405	0.297	22.324	1	0.000	4.074	2.275	7.295
	Sd	0.663	0.285	5.417	1	0.020	1.941	1.110	3.394
	Ct	2.008	0.301	44.384	1	0.000	7.450	4.126	13.450
	Co	1.778	0.332	28.662	1	0.000	5.921	3.088	11.354
	Mf	3.096	0.645	23.036	1	0.000	22.114	6.245	78.301
	Fl	0.281	0.398	0.498	1	0.480	1.324	0.607	2.888
	Sync	2.066	0.444	21.630	1	0.000	7.893	3.305	18.853
	Alt	−18.954	40,192.97	0.000	1	1.000	0.000	0.000	-
	Constant	−2.249	0.317	50.215	1	0.000	0.106	-	-

^a^ Variables according to step 1: Acc, Sd, Ct, Co, Mf, Fl, Sync, and Alt.

**Table 6 ijerph-19-09172-t006:** Model summary.

Step	−2 Log-Likelihood	Cox & Snell R Square	Nagelkerke R Square
1	341.035 ^a^	0.408	0.585

^a^ Estimation ended in iteration 5 because parameter estimates changed by less than 0.01.

**Table 7 ijerph-19-09172-t007:** Contingency table.

			Age (Years)		Total
			Below 49	Over 49	
“Using digital tools helps prevent depression.”	Total disagreement	Responses	8	27	35
		% In the age category	2.1%	19.1%	6.8%
	Disagreement	Responses	26	53	79
		% In the age category	7.0%	37.6%	15.4%
	Neither agree nor disagree	Responses	111	46	157
		% In the age category	29.8%	32.6%	30.5%
	Agreement	Responses	141	11	152
		% In the age category	37.8%	7.8%	29.6%
	Total agreement	Responses	87	4	91
		% In the age category	23.3%	2.9%	17.7%
Total		Responses	373	141	514
		% In the age category	100%	100%	100%

## Data Availability

The data presented in this study are available online at shorturl.at/inIKM (accessed on 15 July 2022).

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
