# Peer review of "Challenges in the Adoption of eHealth and mHealth for Adult Mental Health Management—Evidence from Romania"

_ijerph, 2022, doi:10.3390/ijerph19159172_

Round 1
Reviewer 1 Report
The authors present a study on the challenges in the adoption of eHealth and mHealth for the management of adult mental health.
Taking the title and abstract as reference, the authors do not address the following topics in this manuscript:
1. What are the policies and best practices considered in the current system vs. the one based on eHealth and mHealth?
2. It is necessary to know why they have chosen as a case study the process of detection and treatment of depression in early stages based on the profile of the consumer?
3. There is no previous study on digital tools for mental health purposes that is related to these results. Does it apply to all eHealth and mHealth tools?
4. The authors argue that adults who use technology for health-related needs: 1) accessibility 2) data security and 3) content. These issues are not explained in the manuscript. For example, security
5. Actually, what are the challenges of implementing digital tools as part of the integrated health management system?
6. What are the opportunities for decision makers?
7. How were the Romanian experts on the use of eHealth and mHealth for the prevention, detection and treatment of mild mental disorders selected?
8. The authors must explain the sample. Why 10 specialists?
9. What are the challenges in the ADOPTION of eHealth and mHealth?
Finally, the references are considered old and not updated:
From the year 2001 (Ref. 9 and 13)
From the year 2005 (Ref. 8)
From the year 2006 (Ref. 11 and 52)
From the year 2007 (Ref. 6)
From the year 2009 (Ref. 2 and 15)
From the year 2010 (Ref. 14 and 23)
Etc.
Reviewer 2 Report
The title of the paper says that the paper is about challenges in the adoption of eHealth and mHealth for adult mental health management, with evidence from Romania.
However, the paper is hard to read as it deals with many issues. At least, in the abstract the paper is presenting two different studies, which is not so related to each other.
They studies is not grounded in or motivated by any previous research, so the research gap is not clear. Producers and consumers, as well as marketing are referred to, but it is unclear how this is related to challenges in adoption of eHealth and mHealth in the paper. The paper is messy, with many different objectives. There also seems to one of the results is more important than the others, why not write a paper about that important result then. One motive is however written in the materials and methods, section 2.1.
The concepts of eHealth and mHealth are not fully defined, at least not how the authors of this paper define those concepts.
I doubt that users of mobile phone over 35 years are using their mobile phones for the most basic purposes, as occasional calls and text messages. I just wonder if they should have written 85 instead of 35. The reference for this statement is also old, as it is published in 2015 the research must has been conducted at least one year, maybe two or three, before.
The paper contains very long sentences here and there, and this is not acceptable. Especially in the section literature review this is the case, for example when five major themes of mHealth applications are defined. It will be almost impossible to read and understand the contents in that way.
However, why should the five major themes of mHealth applications be included at all? Those themes seem to be very thorough and broad, and it could be questions how much those are related to the adoption of eHealth and mHealth for adult mental health management. As the paper should be concentrated on mental health management, there seems to be many other areas included in this paper, so the mental health management is a very little part of the whole. What has for example market strategy to do with mental health management?
The materials and methods section is also messy, and it misses an explanation and motivation on the analyses methods.
The discussion refers to hypotheses, why?
I am not commenting the results, as the paper is messy and include far too many objectives, and as it is not concentrated on mental health management.
Reviewer 3 Report
New ways of connecting physicians and patients have arisen. Technology is playing a crucial role in a rapidly growing competitive health services market and for creating a competitive health management system. At the same time the need for knowledge about implementing policies and best practices into the system is highly demanding. The authors proposed a study based on two points of view:
i) qualitative research to determine the opinions of Romanian experts on the risks and benefits of implementing eHealth and mHealth in the process of detecting and treating depression in the early stages based on the consumer profile and
ii) i quantitative research to identify the fu-ture consumption intentions of adults regarding digital tools for mental health purposes.
The main findings of the research highlight three factors that can increase the chances of adults using technology for health-related needs: 1) accessibility 2) data security, and 3) content.
The authors affirms that the paper enriches the literature by bringing a different perspective on the existing challenges of deploying digital tools as part of the integrated health management system and raising the awareness of opportunities for macro and micro decision-makers.
The contribute is interesting and well written
I propose the following suggestions with a pure academic spirit.
1. Abstract must better summarize the sections. Delete for example “Also, the paper enriches the literature …” from the end and summarize the conclusions
2. Insert a clear purpose
3. Insert the limitations in the discussion
4. Check the figures (resolution for example)
5. Discuss the impact of regulations and digital divide
Round 2
Reviewer 1 Report
The authors have reviewed and corrected all comments and suggestions in the manuscript.
Author Response
Greatly appreciated!
Reviewer 2 Report
Unfortunately, there are some more work to do with this paper, before it could be published.
eHealth and mHealth are somewhat described, but what if people prefer one of the two, or the other, or both, this is not clear.
The title refer to mental health, however, much of the paper is related to health-related needs, and not specifically mental health-related needs. This is also obvious especially in the results section.
The first sentence in the introduction needs a reference. Different concepts are used for describing people, as patients, individuals and consumers. Try to use one concept that fulfills the objective, and not use three different concepts that will make it hard to focus for the reader.
Markets are suddenly used in the first sentence of the second paragraph, without being introduced. What are these markets about? Are there any health markets in Romania? What kind of health markets? You also mention mental health management decisions. What kinds of decisions are those? Who make these decisions? Patients, physicians, psychiatrists or PR personnel?
It is also, through the whole paper, unclear if you focus on mental health or health. You also refer to the most important result, in the introduction, and this is related to health-related needs and not to mental health-related needs.
What has age with the aim of the paper? Also, the reference used when presenting the usage of mobile phones is old. There are also so many more factors related to use, than age. It is a too easy way to just use age. Are older people also more exposed to mental health illness, or do they search for more help from the healthcare? Are the older people those who make the most consumption of mental healthcare?
It is not sound to use the word "extremely" in scientific papers.
You are also referring to other medical areas, as medical emergencies, as well as other areas as transport and roads. Why? You also refer to the need of further research in five areas of healthcare. Why, as your title refers to mental health? Why do you refer to the potential to revolutionize social marketing?
You are repeating some information, in various places in the paper. There are more things that are repeated, but one is the presentation of the groups of interviewees. You are also mixing the concepts, as using surveys in qualitative research, and interviews in quantitative research. The description of the analysis method is missing.
The qualitative results seems to be a description of transcripts, without any analysis. I will suggest to thematize the results in any way. You have not done your work.
I cannot find a reason to present many of the figures in the results, as they are not related to mental health. Some information in the figures are also missing.
The discussion is also more or less a mess, unfortunately. For example, one paragraph is more than one page. You are referring to the sample, but the sample is not clearly described in the methods section.
There are many more details that I could have mentioned, but I think the authors really should think about what is interesting in this paper. It is better to be more focused than to include everything.
I am really sorry to be so critical, but after reading this paper a second time I thought that the authors could have made a better job.
Author Response
Please, see the attachment.
